# Effect of Cycling Cadence on Neuromuscular Function: A Systematic Review of Acute and Chronic Alterations

**DOI:** 10.3390/ijerph18157912

**Published:** 2021-07-26

**Authors:** Adrien Mater, Pierre Clos, Romuald Lepers

**Affiliations:** INSERM UMR1093-CAPS, UFR des Sciences du Sport, Université Bourgogne Franche-Comté, F-21000 Dijon, France; pierre.clos@u-bourgogne.fr (P.C.); romuald.lepers@u-bourgogne.fr (R.L.)

**Keywords:** pedaling rate, pedaling frequency, fatigability, EMG, strength

## Abstract

There is a wide range of cadence available to cyclists to produce power, yet they choose to pedal across a narrow one. While neuromuscular alterations during a pedaling bout at non-preferred cadences were previously reviewed, modifications subsequent to one fatiguing session or training intervention have not been focused on. We performed a systematic literature search of PubMed and Web of Science up to the end of 2020. Thirteen relevant articles were identified, among which eleven focused on fatigability and two on training intervention. Cadences were mainly defined as “low” and “high” compared with a range of freely chosen cadences for given power output. However, the heterogeneity of selected cadences, neuromuscular assessment methodology, and selected population makes the comparison between the studies complicated. Even though cycling at a high cadence and high intensity impaired more neuromuscular function and performance than low-cadence cycling, it remains unclear if cycling cadence plays a role in the onset of fatigue. Research concerning the effect of training at non-preferred cadences on neuromuscular adaptation allows us to encourage the use of various training stimuli but not to say whether a range of cadences favors subsequent neuromuscular performance.

## 1. Introduction

Cycling is a common low-impact activity used for daily traveling, recreational practice, and professional competitive sport but also in rehabilitation programs. It appears that all cyclists spontaneously pedal across a narrow range of cadence. This is intriguing because work production per unit of time (i.e., power output), which is the product between pedaling rate and torque applied to the pedal, could theoretically be achieved using a wide range of cadence. Freely chosen cadences (FCC, or preferred cadence) are usually very close among individuals but influenced by practice level. Indeed, professional cyclists prefer cadences above 90 rpm, while active recreational cyclists rather use cadence around 80 rpm [1].

For a given power output, an upward shift of the pedaling cadence reduces the torque applied to the pedal, and vice versa, affecting the physiological and psychological demand of exercise [2]. Cadences can be considered low or high at a given power output when imposed pedaling frequencies were not included in a range of ±25 rpm relative to FCC, usually adopted during training or competition [3]. Sport scientists have been trying to understand why individuals select a cadence rather than another based on physiological, biomechanical, and perceptual parameters. Studies also described acute alterations induced by imposing a pedaling rate below or above the preferred one. A review summarized studies that focused on participants’ responses to exercise, such as oxygen consumption, joint torque, blood lactate accumulation, muscular activation, and perception of effort [2]. The authors reported that all these factors were specifically affected by pedaling rate. Indeed, it appears that these variables follow a “J-curve” or “U-curve” in which the optimal cadence for blood lactate accumulation and oxygen consumption is lower than FCC. In contrast, for minimized mechanical joint torque, the optimal cadence seems above FCC. Then, preferred cadence seems to minimize perceived exertion and would reflect a trade-off between cadences below FCC, lowering oxygen consumption, and above FCC, minimizing mechanical load and thus the possible subsequent alterations of lower limb neuromuscular function.

Neuromuscular alterations could be the cause of a decrease in performance when the effort is prolonged. Indeed, a review investigated the role of pedaling rate on a time trial or time-to-exhaustion performances in relation to energy expenditure [4]. However, the disparity of exercise characteristics—intensities and durations—did not allow them to conclude about a cadence that would optimize performance. It nonetheless seems that cadence impacts performance, and this could occur through fatigue development. Fatigue is defined as “a disabling symptom in which physical and cognitive function is limited by interactions between performance fatigability and perceived fatigability” [5]. Mechanisms involved in the loss of maximal force are commonly investigated through neuromuscular function with the differentiation between muscular and neural components set below and above the neuromuscular junction, respectively.

Moreover, it is well known that fatigue is dependent on the characteristics of the task. Constant load exercise allows studying the impact of the duration and intensity of the task on neuromuscular function. Neural impairments are exacerbated as exercise duration increases (and the intensity that can be sustained decreases), whereas muscular disturbances are greater at higher intensities (and shorter durations of exercise) [6,7]. However, the effect of cadence on neuromuscular alteration after an acute cycling exercise remains to be clarified.

Interestingly, professional cyclists typically use cadences below FCC when training in order to increase muscle tension and provide resistance training-like adaptations, or above FCC to increase the metabolic demand and work on their pedaling gesture to improve their performance at FCC. While Hansen and Rønnestad [3] reported no evidence for a positive effect of training at low cadence, the authors did not emphasize the effect of cadence on chronic neuromuscular alterations while these could contribute to cycling performance [8]. This systematic review aimed to clarify how the utilization of different cycling cadences affects neuromuscular function (i) following a cycling bout, (ii) throughout a cycling exercise, and (iii) following a training period.

## 2. Materials and Methods

The present review was carried out following the “Preferred Reporting Items for Systematic review and Meta-analyses (PRISMA)” guidelines [9] by one scientist. The article search ended on 31 December 2020, and concerned all articles published since 1929. An advanced search was carried out in all files using key-word formulas on PubMed: “((cycling cadence) OR (Pedaling frequency) OR (Pedaling rate)) AND ((neuromuscular) OR (strength) OR (jump) OR (training))” and on Web of Science: “ALL = ((cycling cadence OR Pedaling frequency OR Pedaling rate) AND (neuromuscular OR jump OR training OR strength OR fatigue)).” The research results were added and filtered in Mendeley software (version 1.19.4, 2008-19). Studies were included if they met the following inclusion criteria: (i) participants performed cycling at different cadences, and (ii) neuromuscular adaptations either induced by one fatiguing exercise session or a training intervention were reported. Articles were excluded if: (i) the text was not written in English or French, (ii) the studies did not focus on cycling exercise, or (iii) the studies did not present data on neuromuscular function. Risk of bias assessment was carried out using the Revised Cochrane Risk of Bias Tool for randomized trials (RoB 2.0) independently by two of the authors, following the guideline. Each study was analyzed throughout the five domains proposed by the tool and described as presenting “low risk”, “some concerns”, or a “high risk” of bias. The two investigators then discussed until they found a consensus upon the risk level. Results were then divided into two distinct sections: (1) adaptations caused by one bout of cycling and (2) chronic adaptations after a long-term intervention training. Moreover, to reduce the chance of missing relevant papers, studies’ references were reviewed in order to find further studies of potential interest.

## 3. Results

A total of 4744 (PubMed: 4065, Web of Science: 1036) articles, including duplicates, were identified using electronic databases. After having filtered articles based on their titles and abstracts, 29 articles were included, among which 14 (12 focused on fatiguing bouts and two on training interventions) remained based on the inclusion criteria (including one conference presentation for which only the abstract was available) (Figure 1).

### 3.1. Risk of Bias

Only one in all 14 included studies evaluated with the RoB 2-tool had a high risk of bias that dealt with an overall high risk of bias. Some concern of bias present in the two first items came from the impossibility to blind participants and personnel from cycling intervention because they had to control their cadence. A similar result in the last items resulted from the fact that no indication was mentioned in studies about the selection of the reported results. Low risk of bias due to missing or measurement of outcome data were scored for all studies except one that performed the pre-test one day before the cycling exercise and not immediately before as advised, which means it scored as high risk (Figure 2 and Figure 3).

### 3.2. Study Characteristics

Table 1 presents the main characteristics of studies that investigated neuromuscular impairments induced by one session at different pedaling cadences. Participants of seven studies were cyclists while others were merely healthy weightlifters or team sport players. Fatiguing bouts of pedaling lasted from ≈4 min—for a time-to-exhaustion—to 1 h at an intensity from ≈35% to 95% of peak power output (PPO) determined during an incremental test. In most studies, pedaling intensities were set as a percentage of PPO [10,11,12,13,14] or of maximal oxygen consumption (VO_2_peak) [15,16,17,18]. One study compared cadences at both the same relative metabolic and mechanical work rate [19] and another at power outputs corresponding to the onset of blood lactate accumulation above 3.5 mmol.L^−1^ of lactate [20]. The utilized cadences ranged from 40 rpm to 110 rpm. A team opted for monitoring a large range of cadences without taking into account FFC [14,15] while all others favored cadences below and above the preferred cadence. In the latter case, cadences were considered as low or high with respect to FCC at the same power [3], except in the works of Beelen and Sargeant [19] and de Araujo Ruas et al. [20] who used the same absolute cadences for all participants.

Given the limited number of studies available on the neuromuscular alterations induced by chronic exposure to imposed low or high cadences, the two studies were presented at the end of Table 1. Kristoffersen et al. [21] and Gergley [22] recruited well-trained master and young moderate-trained cyclists for an intervention of 12 and 9 weeks, respectively. Participants trained twice a week with exercise intensity set from 65 to 82% of maximal heart rate. Cadences ranged from 40 to 90 rpm.

The methodology used to assess neuromuscular function differed between studies. First, studies compared values obtained before and after a cycling exercise or a training period. Maximal voluntary contractions (MVC) coupled with electrical stimulation, which was used in two studies [12,13], make it possible to distinguish between the neural and muscular components of performance fatigability [23]. An assessment with countermovement jump (CMJ), used in two others studies [14,20], provides the possibility to assess neuromuscular function in a more practical way [24]. Additional assessments were then used, such as the maximal power output during a 25 s cycling sprint [19] or the maximal number of leg press repetitions carried out with a given load [19,21]. Moreover, muscle biopsies served to determine glycogen depletion and distinguish the type of muscle fibers predominantly used during the exercise [18]. Second, using surface electromyography, studies evaluated the level of muscular activation during the exercise [10,11,15,16,17,25].

### 3.3. Main Outcomes

#### 3.3.1. Acute Neuromuscular Alteration

Studies were further distinguished through two main methodologies: evaluating neuromuscular function during and/or after a cycling exercise. Ahlquist et al. [18] first compared the effect of 30 min of cycling at 85% VO_2_peak at 50 or 100 rpm on glycogen depletion. They found that cadence did not affect glycogen depletion in type I fibers while cycling at 50 rpm led to a greater glycogen depletion in type II fibers than cycling at 100 rpm. Cycling at 80% PPO for 30 min led to a loss of isometric and concentric (up to 120°.s^−1^) knee extensors maximal voluntary contraction force (MVC) without a cadence effect when the pedaling rate was fixed at ±20% FCC [12]. This finding was similar when cycling was performed at 65% PPO for 30 min at 50 or 110 rpm, corresponding to −43% and +25% FCC [13]. Additionally, the latter study found that isometric knee flexors MVC decreased after pedaling at both low and high cadences but not at the preferred one. Moreover, these two studies applied percutaneous electrical stimulation on the femoral nerve to distinguish between muscular and neural components of force production failure. The maximal voluntary activation level decreased in both studies of Lepers et al. [12] and Sarre et al. [13] without difference between cadences. However, changes in the root mean square (RMS) of the electromyographic signal (EMG) during an MVC divided by the maximal muscle compound (M_MAX_) amplitude—EMG_MAX_/M_MAX_ ratio—which is used as a marker of neural drive, showed discrepancies between studies. This ratio decreased for the vastus lateralis (VL) and vastus medialis (VM) muscles after cycling at low and preferred cadences in the study of Lepers et al. [12]. It did not change at any cadences for the VM muscle and decreased for the VL and the rectus femoris (RF) muscles after high pedaling rate cycling in the study of Sarre et al. [13]. In both studies, the maximal torque evoked by motor nerve stimulation at rest decreased whatever the pedaling cadence. M_MAX_ amplitudes were reduced at FCC and low cadence in the study of Sarre et al. [13] but remained unchanged in that of Lepers et al. [12].

Neuromuscular function was also evaluated through high muscle coordination movements such as 25 s cycling sprints. Maximal power output during sprint and so-called rate of fatigue (decrement of power throughout the sprint) were affected by cycling 6 min at 90% VO_2_peak without cadence effect when the pedaling rate was set at 60 and 120 rpm; power output at the high cadence represented 81% of that at the low cadence [19]. Nonetheless, when performed at an equal power output (236 ± 30 W), there was no difference in the peak of power output, but a greater power output decrement during the sprint occurred after cycling at a cadence of 120 compared with 60 rpm. Furthermore, modulating pedaling cadence yielded heterogeneous countermovement jump (CMJ) results. When it was performed after cycling at 35% PPO for 15 min, maximal CMJ height decreased immediately after bouts at FCC −20% and FCC, and returned to pre-exercise values after 1 min of rest, while it remained constant after cycling at FCC +20% [14]. However, when cycling exercise was performed at power outputs corresponding to the onset of blood lactate accumulation (82.5 and 71.9% PPO at 50 and 100 rpm, respectively), the subsequent average height of 10 CMJs was unaltered whatever the pedaling cadence [20]. The latter authors also reported fewer leg press repetitions after cycling at 100 rpm than the control or 50 rpm. Only one study examined the time-to-exhaustion at 95% PPO for different cadences [11]. It showed that FCC −20% yielded longer exercise durations than FCC +20%, while no differences were noticed with FCC.

A second way to assess modulations of neuromuscular function was the use of EMG during cycling. Bessot et al. [11] showed an increase in the EMG RMS of the VM muscle without difference for cadences corresponding to ±20% FCC during exercise. Conversely, both Sarre and Lepers [10] and Vercruyssen et al. [17] found that during 1 h of cycling at 65% PPO and 6 min at 65% VO_2_peak, the EMG RMS of the VL and RF muscles as well as the integrated EMG of the VL and VM muscles raised with time at 110 and 100 rpm, respectively. Moreover, both studies conducted by Takaishi [15,16], which focused on the slope of integrated EMG drift of the VL muscle during 15 min of cycling at 75% and 85% VO_2_peak, reported a quadratic curve with the lowest values at 70 and 80 rpm, respectively. Bessot et al. [25] also monitored the EMG RMS slope of the VM muscle during 21 min of cycling at 65% PPO. They found that the EMG RMS slope at 105 rpm was greater than at 75 rpm, and greater at 60 than 75 and 90 rpm. They also found that the lowest value of the mean quadratic regression was 80 ± 7 rpm, and did not differ from FCC. Additionally, two studies assessed the EMG of the biceps femoris. Bessot et al. [11] found a greater increase at 108 rpm than 72 rpm, while Sarre and Lepers [10] indicated a fall at 50 rpm only. Finally, EMG mean power frequency did not change in the VL, VM, RF, and gastrocnemius lateralis muscles [10,16] but increased for the biceps femoris muscle at all cadences [10].

#### 3.3.2. Neuromuscular Adaptations Following a Training Period

The two interventions retained used different methodologies. Kristoffersen et al. [22] compared two groups of cyclists who added two 90-min sessions per week to their habitual training content. While the control group performed the additional training at moderate intensity (i.e., 73–85% of maximal heart rate) and FCC, the other group performed interval training (i.e., 5 × 6 min with 3 min of recovery) at the same relative intensity but at a cadence of 40 rpm. None of these interventions improved maximal strength assessed with leg extension and leg press movement. Gergley [22] compared two concurrent training programs with the same resistance training content but comprising cycling exercises that differed in terms of cadences (70 rpm or 90 rpm). They found that only the group performing concurrent training while cycling at 70 rpm improved its maximal leg press strength.

## 4. Discussion

This review aimed to summarize neuromuscular alterations following one bout of cycling or repetitive exposure to cycling performed at an imposed cadence, considered as low or high, primarily compared with the preferred one. Because of the heterogeneity of the variables measured and exercise characteristics (i.e., cadence, intensity, duration, and comparison criteria between cadences), the influence of pedaling cadence on neuromuscular function remains elusive yet offers perspectives for future research.

### 4.1. Methodological Considerations

Several precautions must be taken in the present review because of the relative heterogeneity of cadence for both acute and chronic interventions. Firstly, cycling exercises were always performed on a cycle ergometer that excluded contextual consideration, such as the effect of road gradient on cadence [26,27]. Then, the influence of the fitness level and sports background was limited because the freely chosen cadence (FCC) remains consistent within an individual [28], and all conditions (e.g., low cadence) were based on it. The most complicated factor to consider may be exercise intensity. Indeed, it is well known that FCC is intensity-dependent [29] and that, for a given percentage of peak oxygen consumption, shifting from one cadence to another affects power output. Consequently, the conclusions from studies testing the effect of pedaling cadences based on different intensity criteria (e.g., given power output or oxygen uptake) should be compared with caution [29]. Moreover, the cadences considered as low or high are heterogeneous because some investigators chose absolute and other relative (e.g., ±20% FCC) cadences below and above the preferred one [11,12]. For instance, when comparing cadences such as 110 or 50 rpm with FCC, the “high cadence” was 25% above FCC, whereas the “low” one was 43% below FCC [10]. Therefore, cadences considered as low or high can reside within the range of preferred cadences and thus may not affect neuromuscular function distinctly from FCC.

Based on the revised Cochrane assessment method, most studies present an unclear risk of bias. Only one study [20] exhibited a high risk of bias due to baseline measurements having been performed on a separate day and the time delay to perform the tests after the end of the cycling exercise. In addition, biases due to deviations from the intended interventions appear hard to control using such paradigms. Indeed, interventions could not be blind to participants because they were the ones who had to maintain the requested cadences.

### 4.2. Performance Fatigability

Cycling exercise can affect neuromuscular performance such as maximal voluntary isometric force [24], and a shift from the preferred pedaling cadence could exacerbate this phenomenon. Millet and Lepers [23] hypothesized that a shift from the preferred cadence could alter the recruitment of motor units and thus cause a greater maximal voluntary force decrease. This hypothesis came after Ahlquist et al. [18] found that a low-cadence cycling exercise induced a greater glycogen depletion of type II fibers than a higher pedaling rate. Their results suggested that the force applied on the pedals determines the recruitment of motor units. However, while no real consensus emerges from the presently reviewed studies, it appears that when participants were regular cyclists, modulating cadence did not impact neuromuscular performance as expected. Indeed, some authors suggested that trained cyclists can adjust cadences within a range near those usually used during training sessions and competitions without exhibiting more performance fatigability [12]. Moreover, two studies tested performance fatigability and neuromuscular function through isometric contractions, which could likely hide possible alterations at other knee angles or during a dynamic contraction [10,12]. Indeed, Clos et al. [30] showed that a subsequent isometric evaluation did not reflect differences in dynamic torque losses induced by eccentric and concentric tasks. To avoid this limitation, three studies performed dynamic assessments of neuromuscular performance [14,19,20]. Beelen and Sargeant [19] used a sprint on a cycle ergometer before and after fatiguing cycling exercises and, in addition to cycling sprints, de Araujo Ruas et al. [20] used more functional movements such as leg press repetitions with a load corresponding to 10 maximum repetitions and 10 countermovement jumps. Beelen and Sargeant [19] found greater fatigability (i.e., decrease in peak and average power output during the sprint)—after high- than low-cadence cycling at the same work rate. Interestingly, de Araujo Ruas et al. [20] also found a decrease in the number of leg press repetitions after pedaling at a high cadence and not after pedaling at a low cadence performed at a greater power output. These results seem to be in favor of a greater fatigability with high- than low-cadence cycling exercises in non-cyclists. Lastly, Marquez et al. [14] used an exercise intensity and duration typically used for warm-up (15 min at 35% PPO). Nonetheless, jump performance fell directly after cycling at low and preferred cadence, yet returned to baseline after 1 min of rest. Although this result differs from those of the two previously cited studies, it seems that when exercise intensity is sufficient, only high-cadence exercises alter dynamic neuromuscular performance such as cycling sprints and jumps in untrained cyclists.

Other makers of performance fatigability include impaired time trial or a time-to-exhaustion performance [5]. Findings from Bessot et al. [11] allowed us to suppose no clear effect of cadence because time-to-exhaustion was longer at low- compared with high-cadence cycling, but not different from FCC in any condition. A previous literature review supported the fact that cycling performance was altered by cadences higher than the preferred one [31], even if one study found a greater time-to-exhaustion duration at FFC compared with low cadence (50 rpm, without testing other cadences) [32].

### 4.3. Acute Neuromuscular Alterations

An upward drift of EMG RMS during a sustained task is commonly accepted as a marker of impaired muscle ability to produce power as additional motor units are probably recruited and/or firing rate increased despite a steady power output [33]. It appears that the cycling cadences used by all the studies included in this review induced an increase in central drive towards the knee extensors muscles, except the low cadence in the study of Vercruyssen et al. [17], and the low and preferred cadences in the study of Sarre and Lepers [10], where EMG remained constant. Divergences in the nature of cycling exercises (exhausting or not) and gaps between cadences used could lead to a misleading comparison of studies. However, muscular alterations and a decrease in maximal voluntary activation level after the exercise were not influenced by cadence [13,34]. A rise in knee extensor EMG RMS during the exercise was not related to subsequent muscular alterations. However, central drive (EMG_MAX_/M_MAX_ ratio) decreased after cycling at high cadence only [13], which could be explained by a compensatory increase in the neural drive (i.e., EMG RMS) during the exercise, affecting the ability of supraspinal centers to drive the muscle. This is nonetheless speculatory, and it should be noted that changes in the maximal neural drive after cycling did not mirror changes in maximal voluntary activation (i.e., torque). Despite some discrepancies, the results suggest that pedaling at FCC minimizes the rise in EMG RMS throughout a cycling task. Findings from dynamic neuromuscular evaluations after the exercise suggest that non-preferred cadences impact neuromuscular function. These results were supported by the lower increase in EMG RMS at preferred cadences in both studies of Takaishi et al. [15,16] and one from Bessot et al. [25]. Of note, differences in FCC between these studies could be explained by the greater expertise of cyclists and power outputs selected in the second one [35].

The fall in knee flexors EMG RMS found at the low cadence in one study [10] may be explained by a decreased co-activation of this muscle group during the leg extension phase, allowing for a reduced knee extensors work. On the other hand, an increase in the activation of the biceps femoris at high cadence [11] might improve transition phases (at the end of downstroke and during the upstroke of the pedaling movement). It must be noted that despite distinct changes in knee flexors activation throughout low cadence cycling, both studies [10,11] tested trained cyclists. Then, this discrepancy may reflect different individual strategies prevailing in each one of the two moderate sample sizes (n = 11).

Complementary results concerning fiber recruitment patterns could be assessed with EMG methods. A rise in mean power frequency EMG during sustained task suggests recruiting additional muscle fibers and likely type II fibers [36]. Results indicate that motor unit recruitment during cycling did not change with cadence for most of the considered muscles (VL, VM, RF, and gastrocnemius lateralis) except for the biceps femoris, which showed an increased mean power frequency whatever the cadence used [10]. The confidence in the type of motor unit activated while pedaling could likely be improved using high-density EMG [37].

### 4.4. Training

Hansen and Rønnestad [3] already reviewed articles focusing on the effect of a training period at imposed cadences on cycling performance factors such as maximal power output and oxygen consumption or gross efficiency. Although muscular strength was considered as a performance factor and has mainly been focused on training for cyclists [8], only one study (out of seven in Ronnestad’s review) considered it as the main outcome [21]. Then, we reported only one complete study and a conference paper that compared the effect of preferred and low cadences on lower limb muscle strength. It appears that, although pedaling at 40 rpm must be considered as training at a low cadence, no effect was denoted on strength performance assessed with leg press and leg extension [21]. These results could be explained by the relatively low force development induced by sub-maximal low cadence cycling compared with heavy strength training, which is closer to maximal lower limb force capacity. Indeed, the low-cadence exercises used in the studies of Gergley [22] and Kristoffersen et al. [21] were performed over long durations that were closer to endurance than resistance training efforts. Similarly, Koninckx et al. [38] compared 12 weeks of maximal cycling at a relatively low cadence for a sprint—80 rpm for about 825 W over 12 pedal revolutions—with resistance training. While the authors found no improvement in maximal isometric strength, the maximal power output during a 5 s sprint increased after both low-cadence and resistance training periods. However, despite a high torque applied on pedals, maximal sprint cycling training at low cadence did not affect isometric MVC. This result strengthens the point that, when possible, assessments of neuromuscular performance should be realized through functional tests (e.g., cycling sprint, jumping) and/or corresponding to a training regime. Finally, when coupled with resistance training, cycling sessions at a low cadence allowed for greater improvements in lower limb muscle strength than when pedaling at FCC [22]. As the results from training studies that modulated pedaling cadence are scarce, there is a need to multiply such investigations and broaden the training regimens to determine one or several optimal methods [3].

## 5. Conclusions

This review highlights that the role of cycling cadence in performance fatigability and neuromuscular alterations is unclear. It seems that a high cadence at a sufficiently high exercise intensity impairs dynamic neuromuscular performance more than low or preferred cadences, at least in untrained cyclists. One practical consequence of this is that inexperienced cyclists should probably not pedal above their spontaneously chosen cadence if their workout comprises subsequent explosive exercises. Above all, the findings show the relevance of using specific or functional tests for fatigability assessment and of paying attention to the selected population when comparing the impact of pedaling rates. Although research on the effect of cadence during a training period on neuromuscular function is still lacking, it seems essential for coaches to multiply/diversify training regimes even if it means leaving the strict framework of training on the bike. In this perspective, it could be interesting to associate so-called sub-maximal cycling strength training with resistance training. Finally, a method known as eccentric pedaling—resisting against the torque produced by an engine [39]—has recently been spreading in rehabilitation centers [40] and makes it possible to train lower limb muscles at significantly higher levels of force than those allowed by concentric pedaling, leading to superior voluntary force gains [41,42].

## Figures and Tables

**Figure 1 ijerph-18-07912-f001:**
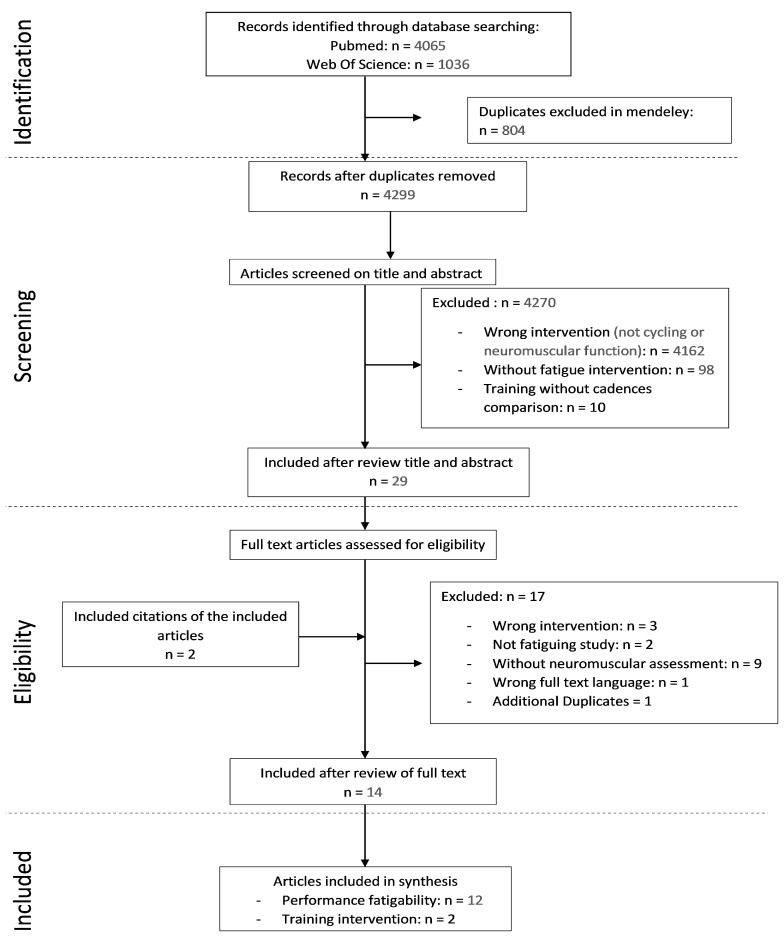
Flow diagram of the reviewing methods based on PRSIMA guidelines.

**Figure 2 ijerph-18-07912-f002:**
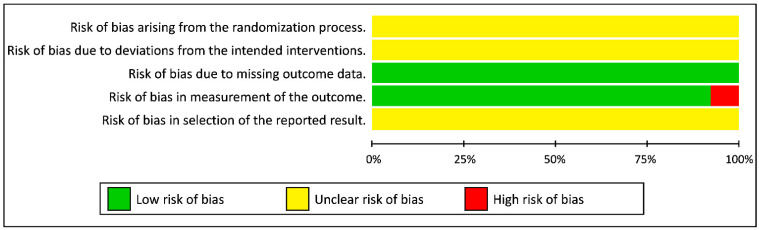
Risk of bias across studies.

**Figure 3 ijerph-18-07912-f003:**
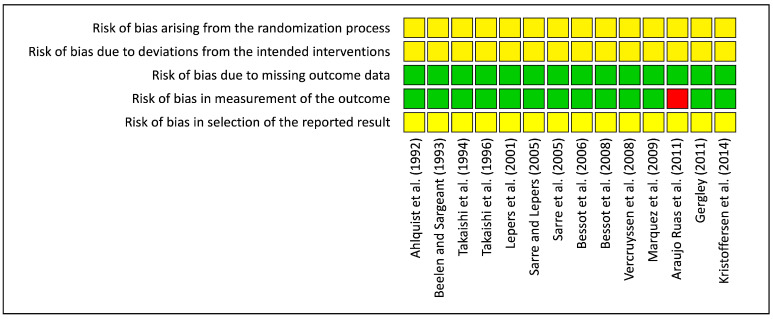
Risk of bias within studies.

**Table 1 ijerph-18-07912-t001:** Effect of pedaling cadence on the acute neuromuscular alteration (FCC: freely chosen cadence, PPO: peak power output, rpm: rotation per minute, EMG: electromyogram, RMS: root mean square, MPF: mean power frequency, MVC: maximal voluntary contraction, CON120 and CON240: concentric contraction at 120 and 240°.s^−1^, ISO: isometric contraction, VL: vastus lateralis, GL: gastrocnemius lateralis, RF: rectus femoris, iEMG: integrated electromyographic activity, HR_max_: maximal heart rate, and RM: repetition maximal).

Study	Participants	Methods	Outcome
**During Cycling Exercise**			
Takaishi et al. (1994)	8 healthy males Age: 20.7 ± 1.5 yrs Mass: 62.5 ± 3.1 kg	15 min at 75% VO_2_peak (from 140 to 210 W) at 40, 50, 60, 70, or 80 rpm Measures: iEMG increase (iEMG slope) in VL during pedaling bout	iEMG followed a quadratic curve with a bottom at about 70 rpm iEMG slope 70 rpm < 50 rpm and 60 rpm, but no differences were found with 40 and 80 rpm
Takaishi et al. (1996)	6 cyclists with 3–4 yrs of road racing experience Age: 20.7 ± 1.5 yrs Mass: 62.5 ± 3.1 kg	15 min at 85% VO_2_peak (from 200 to 240 W) at 50, 60, 70, 80, 90, or 100 rpm Measures: iEMG increase (iEMG slope) in VL during pedaling bout	iEMG slope demonstrated a quadratic curve with bottom near 80 rpm iEMG slope 80 rpm < than other cadences except 90 rpm iEMG slope 90 rpm < than at 100 rpm
Sarre and Lepers, (2005)	11 well-trained male cyclists with at least 4 yrs of racing experience Age: 27.8 ± 5.6 yrs Mass: 71.1 ± 7.8 kg PPO = 382 ± 43 W	60 min at 65% PPO at: FCC (88 ± 11 rpm) 50 rpm 110 rpm Measures: EMG RMS and MPF during pedaling bouts (VL, RF, GL, and BF muscles)	EMG RMS of muscles were differently affected by cadence: EMG RMS of VL and RF ↑ with time at 110 rpm only EMG RMS of BF ↓ at 50 rpm EMG MPF of VL, RF, GL did not change EMG MPF of BF ↑ whatever the cadence
Bessot et al. (2006)	11 male cyclists with 6.5 ± 1.7 yrs of racing experience and 9.8 ± 2.2 h of training per week Age: 19.1 ± 1.8 yrs Mass: 65.9 ± 6.5 kg	Time to exhaustion at 95% PPO at: FCC +20% (72 rpm) FCC −20% (108 rpm) Measures: EMG RMS increase (EMG slope) in VM and BF during pedaling bouts	Time to exhaustion was greater at FCC −20% than FCC + 20%; no difference between FCC and other cadences EMG RMS of VM ↑ regardless of cadence EMG RMS of BF ↑ FCC +20% > FCC −20%
Bessot et al., (2008)	9 competitive male cyclists with 9.8 ± 2.2 h of training per week Age: 21.4 ± 0.7 yrs Mass: 69.6 ± 6.8 kg PPO: 322 ± 32 W	21 min at 65% PPO FCC (86 ± 13 rpm) and 60, 75, 90, 105 rpm Measures: EMG RMS increase (EMG slope) in VM during pedaling bout	EMG slope 105 rpm > than at 75 rpm EMG slope 60 rpm > than at 75 and 90 rpm Optimal cadence to minimize EMG slope determined with regression analysis was 80 ± 7 rpm (not different from FCC)
Vercruyssen et al. (2008)	Well trained male cyclists Age: 25 ± 4 yrs Mass: 76 ± 6 kg VO_2_peak = 64.7 ± 3.1 mL.kg^−1^.min^−1^ PPO = 386 ± 38 W	6 min at 65 ± 7% VO_2_peak at: 50 rpm 100 rpm Measures: iEMG and MPF EMG of VL and VM during pedaling bout	iEMG of VL and VM ↑ during 100 rpm bout only MPF of VL and VM did not change at any cadences
**Pre vs. Post Cycling Exercise**			
Ahlquist et al. (1992)	8 physically active males (4 runners, 4 cyclists) Age: 20–40 yrs Mass: 81 ± 3 kg VO_2_peak = 56.8 mL.kg^−1^.min^−1^	30 min at 85% VO_2_peak (assessed at 75 rpm) at: 50 rpm 100 rpm Measures: muscle biopsy of VL—fiber glycogen depletion	No cadence effect on type I fiber Glycogen depletion 50 rpm > 100 rpm in type II fiber
Beelen and Sargeant (1993)	7 healthy males physically active Age: 27.9 ± 2.7 yrs Mass: 71.0 ± 11.6 kg	Pedaling 6 min at: 60 rpm and 90% VO_2_peak (291 ± 31W) (A) 120 rpm and 90% VO_2_peak (236 ± 30 W) (B) 60 rpm and same workrate as (B) (≈74 ± 11% of VO_2_peak) Measures: 25 s of maximal sprint on cycle ergometer at 60 and 120 rpm	At same VO_2_: ↓ peak power output or kinetic of power output during sprints without cadence effect At same workrate: Decrease in power output over the 25 s after bout at 120 rpm > 60 rpm
Lepers et al. (2001)	11 well-trained male cyclists with at least 4 yrs of racing experience Age: 28 ± 2 yrs Mass: 74 ± 5 kg Height = 183 ± 5 cm PPO = 384 ± 31 W VO_2_peak = 64.1 ± 4.5 mL.kg^−1^.min^−1^	30 min of cycling at 80% of PPO at: FCC (86 ± 4 rpm) FCC +20% (103 ± 5 rpm) FCC −20% (69 ± 3 rpm) Measures: Neuromuscular function of knee extensors muscles	MVC ISO and MVC CON120 ↓ without cadence effect MVC CON240 did not change at any cadence Voluntary activation level ↓ without cadence effect Mechanical evoked torque ↓ whatever the cadence M-wave amplitude did not change at any cadences
Sarre et al. (2005)	11 well-trained male cyclists with at least 4 yrs of racing experience Age: 27.8 ± 5.6 yrs Mass: 71.1 ± 7.8 kg PPO = 382 ± 43 W	60 min at 65% PPO at: FCC (88 ± 11 rpm) 50 rpm 110 rpm Measures: Neuromuscular function of knee extensors and knee flexors muscles	MVC of knee extensors ↓ without cadence effect MVC of knee flexors ↓ after 50 and 110 rpm pedaling bout VAL ↓ without cadence effect EMG RMS/M-wave amplitude of VL and RF ↓ after the 110-rpm bout No change of EMG RMS/M-wave amplitude of VM Evoked torque ↓ whatever the cadence Area of M-waves of VL and VM ↓ after cycling at 50 rpm and FCC
Marquez et al. (2009)	10 physically team sport player males Age: 21 ± 4yrs Mass: 75 ± 6 kg 9.8 ± 2.2 h of training per week PPO = 310 ± 38 W	15 min of cycling at 35% PPO at: FCC (71 rpm) FCC +20% (57 rpm) FCC −20% (85 rpm) Measures: CMJ before and immediately after pedaling bout	CMJ ↓ directly after bout at FCC and FCC −20% but remain unchanged after FCC +20% CMJ return to baseline after 1min of rest at FCC and FCC −20%
Araujo Ruas et al. (2011)	13 weight lifter males Age: 23.0 ± 3.7 yrs Mass: 77.1 ± 8.8 kg 3 weight lifting sessions per week	30 min at onset of blood lactate accumulation (3.5 mmol.L^−1^) at: 50 rpm (82.5% PPO) 100 rpm (71.9% PPO) Measures: 3 sets of 10 RM leg press or 3 sets of 10 maximal countermovement jump	Leg press repetitions ↓ after 100 rpm compared with control condition and 50 rpm Mean CMJ height for all sets did not differed between condition
**Training Interventions**			
Gergley et al. (2011)	14 young moderately trained males Age: 18–23 yrs	2 groups of concurrent training: 90 rpm (65% HR_max_) + resistance training 70 rpm (65% HR_max_) + resistance training 2 sessions per week during 9 weeks Measures: 1RM leg press	↑ lower body strength in 70 rpm + resistance training group only
Kristoffersen et al. (2014)	22 well trained male veteran cyclists Age: 47 ± 6 yrs Mass: 78 ± 7 kg VO2max: 57.9 ± 3.7 mL kg^−1^ min^−1^	2 groups: 40 rpm—5 × 6 min at a HR of 73–82% HR_max_ measured (total of 91 ± 31 h of training) FCC (about 95 rpm) - (total of 88 ± 34 h of training) 2 sessions per week during 12 weeks Measures: 1RM leg press and leg extension	No significant difference in either 1RM leg press or leg extension

## Data Availability

Not applicable.

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
