# Peer review of "Effect of Cycling Cadence on Neuromuscular Function: A Systematic Review of Acute and Chronic Alterations"

_ijerph, 2021, doi:10.3390/ijerph18157912_

Round 1

Reviewer 1 Report

Unfortunately, it is unclear how this review article contributes to the current literature. The focus could have been stronger, and with less overlap with already published work. In addition, it appears that relevant articles (Hansen et al., 2006; Nesi et al., 2004; Stebbins et al., 2014), merely to mention a few examples, are missing. In addition, the language should be considerably improved. An example is “An upward drift of EMG…” (l. 328). EMG is a method. Therefore, the formulation is meaningless.

REFERENCES

Hansen, E. A., Jensen, K., & Pedersen, P. K. (2006). Performance following prolonged sub-maximal cycling at optimal versus freely chosen pedal rate. Eur J Appl Physiol, 98(3), 227-233. http://www.ncbi.nlm.nih.gov/pubmed/16906415 (In File)

Nesi, X., Bosquet, L., Berthoin, S., Dekerle, J., & Pelayo, P. (2004). Effect of a 15% increase in preferred pedal rate on time to exhaustion during heavy exercise. Can. J. Appl. Physiol, 29(2), 146-156. http://www.ncbi.nlm.nih.gov/pubmed/15064424 (In File)

Stebbins, C. L., Moore, J. L., & Casazza, G. A. (2014). Effects of cadence on aerobic capacity following a prolonged, varied intensity cycling trial. J Sports Sci Med, 13(1), 114-119. (In File)

Reviewer 2 Report

The aim of this review is to clarify how the utilization of different cycling cadences affects neuromuscular function.

Here are some annotations:

Why is it not mentioned in the title that it is a systematic review?

Introduction:

The introduction is short but provides the necessary information.

Materials and methods:

- Line 81. The searching phrase in Pubmed does not specify whether it is in all files or in title/abstract….

  • Why didn't you complete the search in other databases?
  • Line 89, I think you can miss interesting papers in Italian and Spanish in a sport like cycling.

Discussion:

  • Probably due to the great heterogeneity of the methodologies used in the different studies included in the review, it is difficult to make conclusions or statements based on the studies. 
  • The conclusions of the review are insufficient, probably due to the hetereogeneity mentioned above.

Round 2

Reviewer 2 Report

The manuscript has been considerably improved. Congratulations.